# Local Genomic Surveillance of Invasive *Streptococcus pyogenes* in Eastern North Carolina (ENC) in 2022–2023

**DOI:** 10.3390/ijms25158179

**Published:** 2024-07-26

**Authors:** Weihua Huang, John E. Markantonis, Changhong Yin, Joseph R. Pozdol, Kimberly P. Briley, John T. Fallon

**Affiliations:** Department of Pathology and Laboratory Medicine, Brody School of Medicine, Eastern Carolina University, Greenville, NC 27834, USA; markantonisj22@ecu.edu (J.E.M.); yinc20@ecu.edu (C.Y.); jpozdol@stanford.edu (J.R.P.); brileyki@ecu.edu (K.P.B.); fallonj19@ecu.edu (J.T.F.)

**Keywords:** *Streptococcus pyogenes*, Group A *Streptococcus* (GAS), ST28/*emm1*, ST399/*emm77*, insertion sequence (IS) IS*Lgar5*, integrative and conjugative element (ICE) *Tn*5801, whole-genome sequencing (WGS), k-mer-based analysis

## Abstract

The recent increase in Group A *Streptococcus* (GAS) incidences in several countries across Europe and some areas of the Unites States (U.S.) has raised concerns. To understand GAS diversity and prevalence, we conducted a local genomic surveillance in Eastern North Carolina (ENC) in 2022–2023 with 95 isolates and compared its results to those of the existing national genomic surveillance in the U.S. in 2015–2021 with 13,064 isolates. We observed their epidemiological changes before and during the COVID-19 pandemic and detected a unique sub-lineage in ENC among the most common invasive GAS strain, ST28/*emm1*. We further discovered a multiple-copy insertion sequence, IS*Lgar5*, in ST399/*emm77* and its single-copy variants in some other GAS strains. We discovered IS*Lgar5* was linked to a *Tn*5801-like *tet*M-carrying integrative and conjugative element, and its copy number was associated with an *erm*T-carrying pRW35-like plasmid. The dynamic insertions of IS*Lgar5* may play a vital role in genome fitness and adaptation, driving GAS evolution relevant to antimicrobial resistance and potentially GAS virulence.

## 1. Introduction

*Streptococcus pyogenes*, also known as Group A *Streptococcus* (GAS), is a strictly human pathogen causing over 700 million infections and over 500,000 deaths annually worldwide as of 2005 [1]. GAS is among the top ten infectious-disease-related causes of death globally [1]. The Centers for Disease Control and Prevention (CDC) has classified erythromycin-resistant GAS as a “concerning threat”, causing 5400 infections and 450 deaths per year in the United States (U.S.) as of 2017, up from 1300 infections and 160 deaths as of 2010 [2]. In December 2022, the World Health Organization (WHO) reported a surge of invasive GAS (iGAS) infections in several European countries [3], including the United Kingdom [4,5], France [6], The Netherlands [7], Belgium [8], Portugal [9], and Denmark [10]. In February 2023, the CDC also reported a higher level of iGAS infections in some U.S. areas, including Colorado and Minnesota [11]. In response, we investigated iGAS infections locally in Eastern North Carolina (ENC) in 2022–2023 and compared them to U.S. iGAS infections in 2015–2021 that have been whole-genome sequenced under the CDC’s Active Bacterial Core surveillance (ABCs).

As of 8 August 2023, there are 1384 multi-locus sequence types (MLSTs) [12] in the PubMLST typing scheme (https://pubmlst.org/) and 2713 cell surface M protein gene (*emm*) types and subtypes [13] in the U.S. CDC M type-specific sequence database (https://ftp.cdc.gov/pub/infectious_diseases/biotech/emmsequ/) for GAS, illustrating its vast genetic diversity. Despite the recent updated *emm*-typing protocol [14], whole-genome sequencing (WGS) is increasingly used to study GAS epidemiology and evolution, as it provides comprehensive information that includes MLSTs, *emm*-types, antimicrobial genes, virulence factors, and alterations in genome organization. By determining the genetic relationship of isolates, WGS-based genomic surveillance could support the close monitoring and tracking of GAS transmission, assist in outbreak investigations, and help inform vaccine development [15,16].

In our previous genomic surveillance [17], we compared various genomic analysis methods, including single-nucleotide variant (SNV)-based, core- and/or pan-genome-based, and k-mer-based analyses. We found an SNV as a unit was too small to identify structural variants, while a gene as a unit in core-/pan-genome analysis was too large. Also, SNV-based analysis is reference-dependent, whereas gene-based analysis is annotation-dependent. Any missing data or potential errors in the reference or during the gene annotation can introduce bias into the results. As such, SNV-based analysis and gene-based core-/pan-genome analysis often result in a relatively lower resolution. In contrast, k-mer-based analysis is adjustable in size and is neither reference- nor annotation-dependent. With these advantages, we employed k-mer (i.e., 31-mer, unless specified otherwise) analysis in this local GAS genomic surveillance study.

## 2. Results

### 2.1. Local Genomic Surveillance of Streptococcus pyogenes Strains

We conducted a retrospective genomic surveillance of 96 GAS isolates collected in ENC from March 2022 to July 2023, one of which (Spyo57) was found to be a mixture of GAS and *Streptococcus agalactiae* after WGS and was thus excluded from the study. Of the remaining 95 isolates, 54 were invasive, 32 were non-invasive (nGAS), and nine were from urine (uGAS). Invasive GAS was defined as the collection of a GAS isolate from a normally sterile site or from a wound in a patient with necrotizing fasciitis or streptococcal toxic shock syndrome (STSS), according to the standard [18]. Due to the retrospective nature of the study, we did not investigate these cases further to determine whether these uGAS strains were invasive or non-invasive. The median age of these patients was 33 years (ranging from 1 to 84), 45 years for iGAS and 10.5 years for nGAS, consistent with the prevalence of nGAS pharyngitis in children. Additionally, 47/95 (49.5%) were male, with 32/54 (59.3%) having iGAS and 14/32 (43.8%) having nGAS. Blood was the main source of iGAS isolates (31/54, 57.4%). Of 54 iGAS patients, 22 were in the intensive care unit (14 males with a median age of 58.5), and 15 had STSS (11 males with a median age of 61), seven of whom died (six males with a median age of 61). The demographic clinical information of these 95 patients, including the antimicrobial susceptibility test results of 36 isolates (27 iGAS), is summarized in Appendix A. No tested isolate was found to be resistant to penicillin, ceftriaxone, or vancomycin.

From WGS with Illumina short-read sequencing, we obtained a draft genome for each GAS isolate using the Shovill de novo assembly pipeline (https://github.com/tseemann/shovill, accessed on 14 March 2024). The median genome size was 1.8 Mb (ranging from 1.69 to 1.95 Mb), consistent with previous GAS genomes in the National Center for Biotechnology Information (NCBI) public genome database. Their genomic characterization of MLSTs and *emm* types is listed in Appendix A. Among the 17 MLSTs and 16 *emm* types identified, the majority were two of the most common GAS strains, ST28/*emm1* (n = 32, 33.68%) and ST36/*emm12* (n = 33, 34.74%). No specific genotype was found to be significantly associated with iGAS infection (Table 1). By utilizing the kWIP algorithm, a weighted genome-wide k-mer analysis [19], we further investigated the relatedness of these 95 GAS isolates. Their phylogeny tree with hierarchical clustering is shown in Figure 1A, and their multi-dimensional scaling (MDS) plot is shown in Figure 1B, with the progenitor M1 (strain SF370, ST28/*emm1*, AE004092) [20], HKU488 (ST28/*emm1*, M1_global_, CP012045) [21], and HKU360 (ST36/*emm12*, CP009612) used as references. For comparisons, phylogeny trees of these 95 GAS isolates from an SNV-based analysis using the snippy algorithm (https://github.com/tseemann/snippy, accessed on 14 March 2024) and from a core-genome analysis using the roary algorithm [22] are demonstrated in Appendix A.

The genomic characterization of antimicrobial resistance and virulence genes in these draft genomes is demonstrated in Appendix A, respectively. Notably, most of the antimicrobial resistance genes (n = 10) were detected in iGAS, rather than in nGAS (Appendix A), which included tetracycline resistance genes (*tet*M:9 and *tet*T:1), erythromycin and macrolide resistance genes (*msr*D:4, *mef*A:4, *erm*B:3 and *erm*T:6), streptothricin resistance gene *sat*4 (n = 5), kanamycin resistance gene *aph*(*3*′)-IIIa (n = 5), and aminoglycoside resistance gene *ant*(*6*)-Ia (n = 3). Additionally, six virulence genes (*fbp54*, *lmb*, *mf/spd*, *ska*, *slo*, and *speB*) were found in all of the isolates, and two virulence genes (*fbaA* and *smeZ*) were significantly associated with iGAS infection (*p* < 0.05, Fisher’s exact probability test, Appendix A). 

The CDC ABCs is an active laboratory- and population-based surveillance system for invasive bacterial pathogens of public health importance, including iGAS. To compare our local genomic surveillance with the existing national surveillance in the U.S., we analyzed all WGS data of iGAS isolates publicly available in the ABCs’ BioProject PRJNA395240, a total of 13,064 isolates. Notably, the iGAS surveillance data in 2015, 2016, and 2017 were reported previously [23,24]. Nevertheless, Table 2 summarizes the main GAS genotypes each year in the U.S. from 2015 to 2021, including ST399/*emm77,* for a comparison to those in our dataset. Among these, ST28/*emm1* was the most prevalent each year, except in 2021 (n = 17, only 2.73%), and ST36/*emm12* decreased almost each year, from 9.1% (n = 132, the third most common) in 2015 to 4.1% (n = 76, 6th) in 2020, with only six (0.96%) in 2021. The decrease of ST101/*emm89* is also notable, as well as that of ST36/*emm12*. Intriguingly, the COVID-19 pandemic not only reduced iGAS incidences [25] but also changed the pattern of prevalent iGAS genotypes. While ST334/*emm82* was significantly increased in 2020 and 2021 (which was second in both years), ST433/*emm49* became the most common in 2021 (15.27%). In contrast to this national surveillance data, ST28/*emm1* and ST36/*emm12* were much more dominant in our local iGAS data during 2022–2023, with 16/54 (29.6%) and 18/54 (33.3%), respectively, along with an upsurge of ST399/*emm77* (4/95, 4.2%).

### 2.2. Identification of Novel Lineages and Sub-Lineages in ST28/emm1

Due to the global dominance of ST28/*emm1* (serotype M1) in iGAS and recent M1_UK_ outbreaks [4,7,8,9,21], we investigated the relatedness of ST28/*emm1* isolates, including our local 32 isolates and the national 907 isolates in the U.S. from 2018 to 2021. We identified three main lineages, namely M1a, M1b, and M1c (Figure 2A). M1a was the dominant cluster (633/939, 67.4%), including progenitor M1, M1_global_, and our 32 isolates. M1a prevailed in 2019 as well as in our cohort. While the M1b isolates were mainly in 2018 and almost disappeared after 2018, M1c became more prevalent in 2020 and 2021, during the COVID-19 pandemic (Figure 2B).

Looking into the M1a lineage, we further identified five sub-lineages, namely M1a-1–M1a-5. Each sub-lineage contained two subclusters, one harboring an intact prophage 315 (~36 kb, comparable to 1,168,867–1,204,880 in the MGAS2221 strain, CP043530) [26], and the other having deletions in the ~11 kb region of the prophage (comparable to 1,190,373–1,201,034 in CP043530, namely Δ11k) (Figure 2C and Appendix A). Progenitor M1 belonged to M1a-1 with Δ11k; M1_global_ belonged to M1a-2, a dominant sub-lineage (415/633, 65.6%); the majority of our M1 isolates (22/32, 68.8%) belonged to M1a-3, 14 of which were iGAS strains; only 11 isolates belonged to M1a-4, mainly in 2018 and 2019; and four of our M1 isolates were M1a-5, a new sub-lineage in the U.S., two of which were iGAS strains (Figure 2C,D). Notably, our M1 isolates in M1a-1 (n = 1) and M1a-2 (n = 5) were all nGAS strains.

We further investigated the M1a-2 isolates with an intact prophage 315 (n = 364), including our five nGAS strains. We included four additional complete genome references: SP1426 (CP060268) as another M1_global_; and SP1448 (CP060267), SP1380 (CP060269), and SP1384 (CP060270) as M1_UK_ [21]. We uncovered three main sub-lineages, namely M1a-2a, -2b, and -2c (Appendix A). Both M1_global_ and M1_UK_ belonged to M1a-2b, which contained our three nGAS and 79 iGAS strains in the U.S. Despite a recent M1_UK_ outbreak in European countries, M1a-2b in the U.S. was not as prevalent as M1a-2a (n = 226) during 2018–2020.

Our focused kWIP analysis on the M1a-2b isolates further distinguished M1_UK_ from M1_global_ (Appendix A), the discrepancy between which was defined as 27 SNVs and four indels [21,27]. Our three nGAS strains, along with 21 isolates from 2019 and 20 isolates from 2020 in the U.S., were close to M1_UK_. We validated this clustering result by using the SNV-based analysis with MGAS5005 (M1 serotype, NC_007297) [26] as a reference and confirmed 23/27 SNVs in three M1_UK_ isolates and all 27 SNVs in the remaining 41 M1_UK_ isolates (Appendix A). The relatedness clustering of all ST28/*emm1* GAS strains (n = 939) in our local and national comparative genomic analyses is detailed in Appendix A.

### 2.3. Multiple-Copy ISLgar5 and Single-Copy ISLgar5 Variants in Streptococcus pyogenes

To identify genetic elements associated with iGAS infection, we performed a genome-wide association study (GWAS) on 54 iGAS vs. 32 nGAS strains using the kmdiff algorithm, a differential k-mer analysis between two populations [28]. A total of 189 differential k-mers were identified in iGAS with significance (*p* ≤ 2.2 × 10^−9^), from which two fragments (123 bp each) were assembled de novo. A BLAST search of these two fragments hit a single insertion sequence, IS*Lgar5*, with one at its 5′ end and the other at its 3′ end (Figure 3A). According to ISFinder [29] (https://www-is.biotoul.fr/, accessed on 8 August 2023), IS*Lgar5* is 1336 bp long, with 26 imperfect terminal repeat sequences (IRs, four mismatches), belonging to the IS256 family. Initially identified in *Lactococcus garvieae* IPLA31405, IS*Lgar5* has been found in firmicutes, mainly in *Enterococcus*, *Staphylococcus*, and *Streptococcus*, but only one has been found in GAS, i.e., emmSTG866.1 (CP035428) in Kenya in 2005 [16], in our BLAST search of the NCBI public nucleotide database.

However, we later found out that our GWAS was biased by multiple copies (14–15 as estimated) of IS*Lgar5* existing in our four ST399/*emm77* isolates, based upon the assessment of differential sequencing coverage between the mobile genetic element (MGE) and the whole genome. In our screening of the NCBI Sequence Read Archive (SRA) public database, we detected a total of 111 isolates harboring multiple copies of IS*Lgar5*, all of which belonged to the genotype ST399/*emm77*, with one exception, SRR18933625 (ST458/*emm28*) (Appendix A). Of 110 ST399/*emm77* isolates, 99 were from the ABCs’ BioProject PRJNA395240 in the U.S. from 2015 to 2021; one was from Texas in 2014 (PRJNA494557); one was from Ireland in 2020 (PRJEB34287); and nine were from the UK before 2015 (PRJEB13551, PRJEB12015, and PRJEB17673), indicating its geographically wide existence. The estimated IS*Lgar5* abundance in each ST399/*emm77* isolate ranged from 3 to 39, with a median of 11.

Moreover, we found 26 GAS isolates harboring only a single copy of IS*Lgar5*, 22 in the U.S. in 2015–2021 (PRJNA395240), one in the Netherlands in 2022 (PRJNA967239), and three in Norway in 2023 (PRJEB42599) [30] (Appendix A), in addition to the aforementioned emmSTG866.1 (ST450). Intriguingly, most of these single-copy IS*Lgar5* were IS*Lgar5* variants, namely IS*Lgar5*-1, -2, and -3. IS*Lgar5*-1 (n = 5) in emmSTG866.1, two ST181/*emm78*, one ST677/*emm12*, and one ST1057/*emm104* had one to two nucleotide changes, causing a W296R mutation in the IS*Lgar5*-encoded transposase. IS*Lgar5*-2 (n = 10) in seven various genotypes had the same nine nucleotide variants, leading to a transposase mutant with K149N and A291T alterations. IS*Lgar5*-3, unique in nine ST904/*emm77* isolates, had a nucleotide variant in addition to those in IS*Lgar5*-2 with three amino acids being changed in the transposase: K149N, S258R, and A291T (Figure 3B and Appendix A).

### 2.4. Association of ISLgar5 with Streptococcus pyogenes Antimicrobial Resistance

Most IS*Lgar5*-carrying isolates (121/129, 93.8%), including our four ST399/*emm77* isolates, harbored the *tet*M gene, conferring resistance to tetracycline [15,31], which caused us to suspect that IS*Lgar5* is linked to *tet*M. By using the emmSTG866.1 genome as a reference, we identified a novel ~26 kb *Tn*5801-like integrative and conjugative element (ICE, 986,768–960,966 in CP035428) carrying both IS*Lgar5* and *tet*M (Figure 3C). The ICE starts with IS*Lgar5* and ends with a site-specific integrase, between which are genes encoding ATP-dependent endonuclease and helicase, anti-restriction protein ArdA, conjugal transfer proteins, regulatory transcription factors (the XRE family and sigma-70 family), accessory *tet*M, etc. All *tet*M-positive iGAS strains harbored this *tet*M-carrying ICE in the chromosome, whereas the *tet*M-negative ST399/*emm77* isolates (n = 7) had a large deletion of internal genetic components, with only IS*Lgar5*, the sigma-70 family transcription factor, and integrase remaining (Figure 3C). Notably, compared to its genome sequencing coverage, SRR7706789 had a relatively lower sequencing coverage on the ICE, suggesting this MGE was not fully integrated into the genome (either moving in or moving out), putting the *tet*M positivity in question (Appendix A). In the IS*Lgar5*-3-carrying ST904/*emm77* isolate, the *tet*M-carrying ICE had a 53 bp deletion at the 3′ end of a gene encoding a DUF87 domain-containing protein, disrupting its translation (Figure 3C). Mutations of the *Tn*5801-like ICE have also been found in IS*Lgar5*-1-carrying SRR18923745 (ST677/*emm12*) isolates.

In addition to the IS*Lgar5*- and *tet*M-carrying ICE, all four ST399/*emm77* isolates in our GAS cohort had a pRW35 [32]-like plasmid carrying *erm*T, named pRW35-ENC (CP136949/CP136952). Of 99 ST399/*emm77* isolates in PRJNA395240, 75 (75.8%) also had the *erm*T-carrying plasmid, four had chromosomal *erm*B, and one had both *erm*A and *erm*T. These *erm* genes are responsible for GAS resistance to erythromycin, azithromycin, clarithromycin, and clindamycin [15,33]. Interestingly, none of the single-copy IS*Lgar5*-containing GAS strains carried *erm*T and only one of ten ST399/*emm77* isolates before 2015 (ERR1733450 in the UK) carried *erm*T, leading us to suspect that multiple-copy IS*Lgar5* is associated with *erm*T-related antimicrobial resistance and linked to ST399/*emm77* evolution. Figure 3D demonstrates that ST399/*emm77* isolates with *erm*T had a significantly higher copy number of IS*Lgar5* (mean = 12.9) than those without *erm*T (mean = 7.0, *p* = 1.3 × 10^−9^, two-tailed Student’s *t*-test) and that more ST399/*emm77* isolates with *erm*T were identified after 2018 (37/39, 94.9%), compared to 38/60 (63.3%) in 2015–2018 and 1/10 (10%) before 2015.

### 2.5. Characterization of Multiple-Copy ISLgar5 in ST399/emm77

Currently, the complete ST399/*emm77* genome is not available in the NCBI public genome database. We therefore conducted Pacific Bioscience (PacBio) HiFi long-read sequencing on Spyo01 and Spyo09, along with an M1a-3 isolate Spyo06, to obtain their complete genomes. Figure 4A demonstrates a whole-genome comparison of these isolates with references M1, emmSTG866.1, and emm77 (ST588, CP035439). We confirmed 14 copies of IS*Lgar5* in the Spyo01 chromosome (CP136948), but only 13 copies in Spyo09 (CP136951), one copy short of our initial estimation. Their insertion sites and correspondingly affected genes are summarized in Appendix A. Interestingly, IS*Lgar5* had also inserted into its own ICE in both Spyo01 and Spyo09 (#6, contrast to original #5, Figure 3C), which might prevent the MGE from further transposition. Apart from one IS*Lgar5* copy difference between Spyo01 and Spyo09, there was an insertion of 10 bp (2× TGTTT repeat) in Spyo01 impacting the expression of LPXTG-anchored collagen-like adhesin Scl2 (R3H37_05230 in Spyo01, compared to R3H61_05235 in Spyo09) and an insertion of ~5.3 kb in Spyo09 containing eight genes (R3H61_07350–07385), including exotoxin gene *spe*J, which is consistent with the virulence factor profiling shown in Appendix A. Other than these discrepancies, the genomes of Spyo01 and Spyo09 are almost identical.

The insertion of IS*Lgar5* generates an eight bp duplicate repeat (DR) or target site repeat (TSR) at its ends. We uncovered that all single-copy IS*Lgar5* isolates and their variants had the same conservative DR, TTATAATG, in the IS*Lgar5*-containing ICE. However, in the ICE of multiple-copy IS*Lgar5*-containing iGAS strains, there was a four or five bp deletion in their 3′ DR. Moreover, we found more deletions of various lengths at the 3′ end of IS*Lgar5*, e.g., 82 bp at #7, 598 bp at #13, and 5994 bp at #10, including the aforementioned large deletion in the *tet*M-negative ST399/*emm77* isolates, which led to their 3′ DR differing from their 5′ DR (Appendix A). We thus suspect that multiple-copy IS*Lgar5* variants may have an additional role in deletion, distinct from single-copy IS*Lgar5* variants.

To investigate the effect of multiple-copy IS*Lgar5* insertions, we compared the genome-wide gene expression of our ST399/*emm77*, M1a-3 iGAS, and nGAS isolates (n = four each) via RNA sequencing (RNA-seq). We observed little difference in gene expression between M1a-3 iGAS and nGAS using Spyo06 (CP136950) as a reference but significant changes between ST399/*emm77* and ST28/*emm1* using the Spyo09 genome (CP136951-CP136952) as a reference (Figure 4B–D). In total, 39 genes were identified with significant differential expression (adjusted *p*-value ≤ 0.05, |log_2_FoldChange| ≥ 1): 14 down- and 25 up-regulated in ST399/*emm77* (Appendix A). It is no surprise that M-related protein (Mrp) and its clustered-together YSIRK-type signal peptide-containing protein (R3H61_08280) were expressed in ST399/*emm77*, replacing the M protein in ST28/*emm1*. Compared to ST28/*emm1*, ST399/*emm77* had a reduced expression of exotoxin SpeB but an increased expression of three LPXTG motif-containing cell-wall-anchored proteins, R3H61_08345, and adhesins Scl1 and Scl2, which might modulate GAS virulence and contribute to GAS strains’ adhesion, colonization, and disease-causing abilities [34].

## 3. Discussion

We employed a k-mer-based kWIP relatedness analysis for the genomic surveillance of local iGAS strains and compared the results to those of the national surveillance. The k-mer-based population approach has a discriminatory power to differentiate isolates via mainly structural variants (e.g., Δ11k and contaminated phi174), as well as SNVs (e.g., M1_UK_ vs. M1_global_). In the national genomic re-analysis of M1 GAS, we found three main lineages and several sub-lineages and uncovered GAS epidemiological changes in the U.S. before and during the COVID-19 pandemic. Notably, compared to previous M1_UK_ emergence [27] and recent M1_UK_ outbreaks in European countries [4,7,8,9,21], only a limited number (n = 41, 2018–2020) of M1_UK_ isolates were identified in the U.S., similar to an earlier observation [35], which demonstrates a geographic difference in GAS circulation. Additionally, we revealed significant GAS prevalence differences between 2018 and 2019: the M1b lineage dominated in 2018 but disappeared from 2019 to 2020, and the M1a lineage prevailed in 2019. During the COVID-19 pandemic, the iGAS cases and deaths were significantly reduced [25], from 7.6 in 2019 to 6.1 in 2020 and from 0.67 in 2019 to 0.55 in 2020 per 100,000 population, respectively (https://www.cdc.gov/abcs/reports-findings/surv-reports.html, accessed on 14 May 2024). As shown in Table 2, we also observed a pattern change in iGAS prevalence. These phenomenal changes might be closely correlated with the implementation of COVID-19-associated nonpharmaceutical interventions [25].

Our local genomic surveillance of GAS revealed a unique sub-lineage, M1a-5, in ENC among the most common iGAS ST28/*emm1* isolate. However, our local genomic surveillance was conducted in a relatively short time period with a relatively small size and was confined to ENC, a relatively large geographic region with a relatively small population. The significant age difference between our iGAS and nGAS collection is noted, mainly because nGAS is much more common among young people, whereas iGAS is much more common among adults. Compared to national GAS in the U.S. before 2022, our local GAS in 2022–2023 showed a high percentage of ST28/*emm1*, ST36/*emm12*, and ST399/*emm77* isolates. Unfortunately, to date, there are no national data available on iGAS in 2022–2023. It will be of additional interest in the future to compare our local GAS data with the national GAS data in the same period to further explore GAS geographic features.

We noticed there is an inconsistency between resistance phenotypes and genotypes in the Spyo02, Spyo11, Spyo19, Spyo35, and Spyo51 strains. We therefore investigated rare erythromycin and clindamycin resistance mechanisms of mutations in 23S rRNA, as well as ribosomal proteins L4 and L22 [36]. Although we observed some mutations in 23S rRNA, e.g., A1302C, T2021G, and T2166C (using AE004092 as a reference), they demonstrated no significant difference between our susceptible and resistant isolates, except a rare mutation, C2702T, in Spyo11. No mutation in L4 or L22 was found. We reason that additional resistance mechanism(s) might exist in these erythromycin- and clindamycin-resistant isolates, which remain to be explored.

We employed both a kWIP k-mer-based relatedness analysis and a kmdiff differential analysis to investigate ST28/*emm1*. The combined use of kWIP and kmdiff can discriminate clusters and/or subclusters without any reference or genome annotation and can identify structural variants, such as Δ11k, phi174 contamination, and IS*Lgar5*. In a subsequent global survey, we found multiple-copy IS*Lgar5* in ST399/*emm77* isolates and single-copy IS*Lgar5* variants in other GAS strains. The host specificity of multiple-copy IS*Lgar5* in ST399/*emm77* (except SRR18933625) strains and single-copy IS*Lgar5*-3 in ST904/*emm77* strains is intriguing. Based upon our current characterization of IS*Lgar5* and its variants, we assume the IS*Lgar5* mobility between GAS strains is through the IS*Lgar5*- and *tet*M-carrying ICE, whereas the IS*Lgar5* mobility inside GAS depends on the T980 (W296) variant.

The IS*Lgar5*- and *tet*M-carrying *Tn*5801-like ICE was originally detected in *Staphylococcus aureus* Mu50 (NC_002758) [37], and its presence was documented as early as 1953 in *Streptococcus agalactiae* [38]. Compared to *Tn*916 [39,40], *Tn*5801 has the common synteny but with an extra IS*Lgar5* transposase, ATP-dependent endonuclease, and ATP-dependent helicase at one end, and it differs in the site-specific integrase at the other end. In consideration of ICE excision circularization, we propose a four-component cluster for the transfer mechanism of *Tn*5801-like ICE, comparable to the bacterial UvrABCD excision repair system [41]. Interestingly, in a genomic regional comparison between CP035428 and CP043530, this ICE integration generated a 11 bp target site replication (though *att*P GAGTGGGAGTA and *att*B GAATGGGAATA were not a perfect match), also comparable to the 12 bp excision by UvrABC. The 11 bp sequence located at the end of the glutamine-hydrolyzing GMP synthase-coding gene *gua*A is consistently found to be in association with *Tn*5801-like MGE in various species of *Enterococcus*, *Staphylococcus*, and *Streptococcus* [42], suggesting it is the Chi (crossover hotspot instigator) site specific for the *Tn*5801-like tyrosine recombinase.

IS*Lgar5* contains a unique 396 aa transposase with a DDE motif that has an 88% similarity to that in ISEfm2, both of which belong to the IS256 family. The IS256 family is featured with replicative copy-out/paste-in transposition, generating an eight to nine bp DR on insertion [43,44]. The deletions identified at the 3′-end of multiple-copy IS*Lgar5* are extraordinary. Essential for multiple-copy IS*Lgar5* and its deletions, W296 resides on the protein surface away from the DDE activity center in our predicted three-dimensional protein structure. The vital role of W296 in IS*Lgar5* transposition thus merits further investigation.

Like multiple-copy IS*Lgar5*, IS256 also produces multiple insertions throughout the bacterial chromosome [45]. A recent study [46] demonstrated that IS256 mobility was tightly controlled at the transcriptional level and that IS256 insertion abundance coincided with phage infection and antibiotic exposure. Our genomic surveillance also revealed the association of IS*Lgar5* with antimicrobial resistance. Additionally, our transcriptomics analysis suggested that multiple-copy IS*Lgar5* might alter GAS virulence and pathogenesis. These results support that multiple-copy IS*Lgar5* is a crucial component for ST399/*emm77* rapid genome fitness and adaptation. Environmental selective pressure, such as phages and antibiotics, may promote IS*Lgar5* diversification inside the GAS genome. After all, insertion sequences are key drivers of bacterial genome evolution, shaping bacterial responses [43,46,47].

In conclusion, using k-mer-based analyses, we revealed three main clusters of M1, i.e., M1a, M1b, and M1c, and a novel sub-cluster, i.e., M1a-5, in our local genomic surveillance of GAS along with the national genomic surveillance of iGAS that is publicly available. We also identified multiple-copy IS*Lgar5* specific in ST399/*emm77* isolates*,* the copy number of which was associated with antibiotic resistance, i.e., *erm*T. Meanwhile, we demonstrated the power of reference-free and annotation-free genomic analyses, which should have wider applications in the future.

## 4. Materials and Methods

### 4.1. Bacterial Isolates

All GAS isolates in this retrospective study were collected from March 2022 to July 2023 in the Clinical Microbiology Laboratory of the ECU Health Medical Center. Clinical data on all cases were retrieved from patients’ electronic medical records for review and consisted of age, gender, sample collection source and date, infection type/site, clinical diagnosis, and mortality. The clinically isolated GAS strains were cultured on 5% sheep blood agar plates and amplified in LIM broth (Todd Hewitt with CAN) or Tryptic Soy Broth (Becton, Dickinson and Company, Sparks, MD, USA). When clinically indicated, antimicrobial reagent susceptibility testing was performed utilizing a combination of ETEST^®^ for penicillin, ceftriaxone, and vancomycin (bioMérieux Inc., Durham, NC, USA) and Kirby-Bauer disk diffusion for erythromycin and clindamycin (Becton, Dickinson and Company). This retrospective study was approved by the University and Medical Center Institutional Review Board at East Carolina University (UMCIRB 23-000323).

### 4.2. Short-Read Whole-Genome Sequencing

Genomic DNA was extracted using the GenFind v3 kit and the Biomek i7 automation system (Beckman Coulter, Indianapolis, IN, USA). DNA quantification was performed using the AccuClear Nano dsDNA Assay and SpectraMax iD3 Fluorometer (Molecular Devices, San Jose, CA, USA). Multiplex sequencing libraries were prepared with the Nextera XT Library Prep kit (Illumina, San Diego, CA, USA), the quality and quantity of which were measured using the 4200 TapeStation (Agilent, Santa Clara, CA, USA) and the Qubit 4 Fluorometer (ThermoFisher, Waltham, MA, USA), respectively. Paired-end sequencing (300 × 2 cycles) was conducted using the NextSeq 2000 or MiSeq (Illumina) platform.

### 4.3. Long-Read Whole-Genome Sequencing

High-molecular-weight (HMW) genomic DNA was extracted using the Quick-DNA HMW Magabead kit (Zymo Research, Irvine, CA, USA) and sheared to 7–10 kb using g-TUBE (Covaris, Woburn, MA, USA). A HiFi long-read sequencing library of each isolate was prepared using the SMRTbell Prep kit 3.0 and Barcoded Adapter Plate 3.0 (PacBio, Menlo Park, CA, USA). Pooled libraries were loaded using the Polymerase Binding kit 3.2 and sequenced with an SMRTCell 8M tray in the Sequel IIe system (PacBio), with 2 h pre-extension time and 30 h movie run.

### 4.4. RNA Sequencing (RNA-Seq)

Four ST399/*emm77* strains (Spyo01, Spyo09, Spyo10, and Spyo16), four ST28/*emm1* iGAS strains (Spyo06, Spyo35, Spyo37, and Spyo43), and four ST28/*emm1* nGAS strains (Spyo61, Spyo65, Spyo67, and Spyo70) were selected for RNA-seq analysis. Bacteria were disrupted in the TRIzol reagent (ThermoFisher) with the Lysing Matrix B using the FastPrep-24 5G instrument (MP Biomedicals, Santa Ana, CA, USA). Total RNA was extracted from each isolate using the Direct-zol RNA Microprep kit (Zymo Research). Ribosome RNA (rRNA) was removed using the Ribo-Zero Bacteria rRNA Depletion kit (Illumina), and multiplex sequencing libraries were prepared using the TruSeq Stranded Total RNA Prep kit (Illumina). Pooled libraries were paired-end sequenced (75 × 2 cycles) in the MiSeq system (Illumina).

### 4.5. Bioinformatics Analysis

Sequencing adaptors and low-quality sequences were removed using Trimmomatic v0.39 [48]. Draft genomes were assembled de novo using SPAdes v3.15.0 [49] with k-mers set at 21, 33, 55, 77, 99, and 127 for initial local and national surveillance and using Shovill (https://github.com/tseemann/shovill, accessed on 14 March 2024) for local refined surveillance. The resulting contigs were subjected to (1) MLST identification using mlst v2.19.0 (https://github.com/tseemann/mlst, accessed on 25 May 2022) against PubMLST typing schemes (https://pubmlst.org/organisms/streptococcus-pyogenes/, accessed on 25 May 2022); (2) antimicrobial resistance and virulence gene identification using ABRicate (https://github.com/tseemann/abricate, accessed on 25 May 2022) against the CARD [50], ResFinder [51], ARG-ANNOT [52], NCBI, and VFDB [53] databases; (3) M protein gene (*emm*) typing using emmtyper (https://github.com/MDU-PHL/emmtyper, accessed on 14 March 2024), with manual inspection of those with more than one *emm* type; (4) alignment- and annotation-free isolate relatedness analysis using the k-mer weighted inner product (kWIP, v0.2.0) [19]; (5) statistical significance of MDS clustering using SigClust-MDS [54]; (6) SNV-based genomic phylogeny analysis using Snippy algorithm (https://github.com/tseemann/snippy, accessed on 14 March 2024); (7) assembled genome annotation using Prokka [55]; (8) pan- and core-genome phylogeny analysis using Roary [22]; (9) phylogeny tree generation and visualization using FastTree [56] and iTOL (https://itol.embl.de/, accessed on 14 March 2024); and (10) GWAS of iGAS (cases) vs. nGAS (controls) using kmdiff differential k-mer analysis [28]. The GWAS resulting k-mers in the cases were assembled using Velvet v1.2.10 [57] with k-mer set at 21; assembled contigs were subsequently BLASTed to the NCBI public nucleotide database for identification of any potential associated gene(s). For the verification of gene existence in each isolate, the BWA aligner v0.7.17-r1188 [58] was used for raw sequence reads alignment and mapping to the identified IS*Lgar5*. In survey of IS*Lgar5*, the NCBI public SRA database was screened, which included BioProjects PRJNA967239 (117 isolates) from the Netherlands in 2021–2022, PRJEB34287 (88) from Ireland in 2020, PRJEB42599 (411) from Norway in 2021–2023, PRJEB13551 (555), PRJEB12015 (1215), and PRJEB17673 (3047) from the UK before 2015, and PRJNA395240 (13,064) from the U.S. in 2015–2021. Complete genomes were hybrid-assembled using SPAdes using both Illumina short-reads and PacBio long-reads, followed by manual inspection and correction with the Integrative Genomics Viewer (IGV, v2.9.1) [59]. Final genomes and plasmid pRW35-ENC were submitted to the NCBI and annotated using the Prokaryotic Genome Annotation Pipeline (PGAP). Whole-genome sequence alignment and visualization were performed using Mauve v2.4.0 [60]. Prophages were detected using PHASTER [61]. For RNA-seq, sequencing reads after trimmomatic cleaning were mapped to the reference genome, Spyo06 (CP136950) or Spyo09 (CP136951-CP136952), using EDGE-pro [62]. Differential gene expression analysis was performed using DESeq2 [63], in which the *p*-values attained by the Wald test were corrected for multiple testing using the Benjamini and Hochberg method.

## Figures and Tables

**Figure 1 ijms-25-08179-f001:**
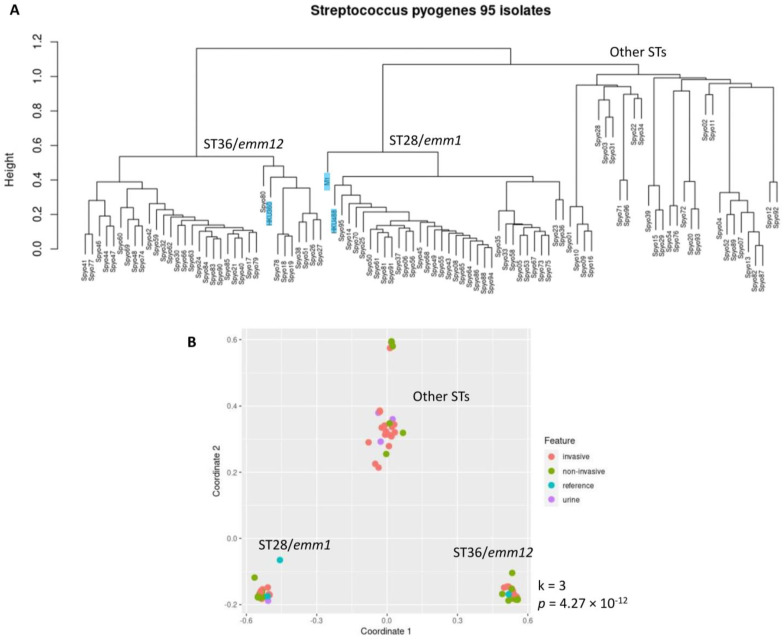
Relatedness analysis of 95 *Streptococcus pyogenes* isolates in Eastern North Carolina, using kWIP analysis. (**A**) Hierarchical clustering of the 95 Group A *Streptococcus* isolates along with three complete genome references: M1 (ST28/*emm1*), HKU488 (ST28/*emm1*, M1_UK_), and HKU360 (ST36/*emm12*). Height indicates the degree of difference between branches. (**B**) Multi-dimensional scaling (MDS) plot of the 95 GAS isolates along with the references. ST28/*emm1* and ST36/*emm12* are the main two clusters. Statistical significance *p*-value of MDS clustering is listed beside the figure, along with cluster number k. ST: sequence type.

**Figure 2 ijms-25-08179-f002:**
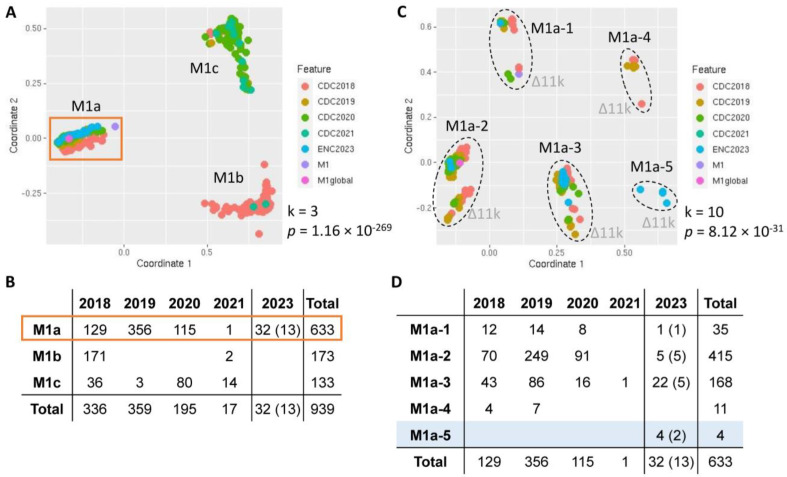
Relatedness analysis of *Streptococcus pyogenes* ST28/*emm1* (serotype M1) isolates in both local and national comparative genomic surveillances, using kWIP analysis. Statistical significance *p*-value of each multi-dimensional scaling (MDS) clustering is listed beside the figure, along with cluster number k. (**A**) MDS plot of 939 M1 isolates. Three main clusters, M1a, M1b, and M1c, are identified, compositions of which are parsed in (**B**), with the dominant M1a isolates boxed in orange. (**C**) MDS plot of 633 M1a isolates demonstrates five different clusters, compositions of which are parsed in (**D**). Each cluster has two sub-clusters, one with intact prophage 315 and the other with deletions in the ~11 kb region of the prophage (Δ11k). Numbers in parentheses (**B**,**D**) are non-invasive isolates sequenced in Eastern North Carolina (ENC). The novel sub-lineage identified in ENC is shaded in blue (**D**), which comprises two invasive and two non-invasive isolates.

**Figure 3 ijms-25-08179-f003:**
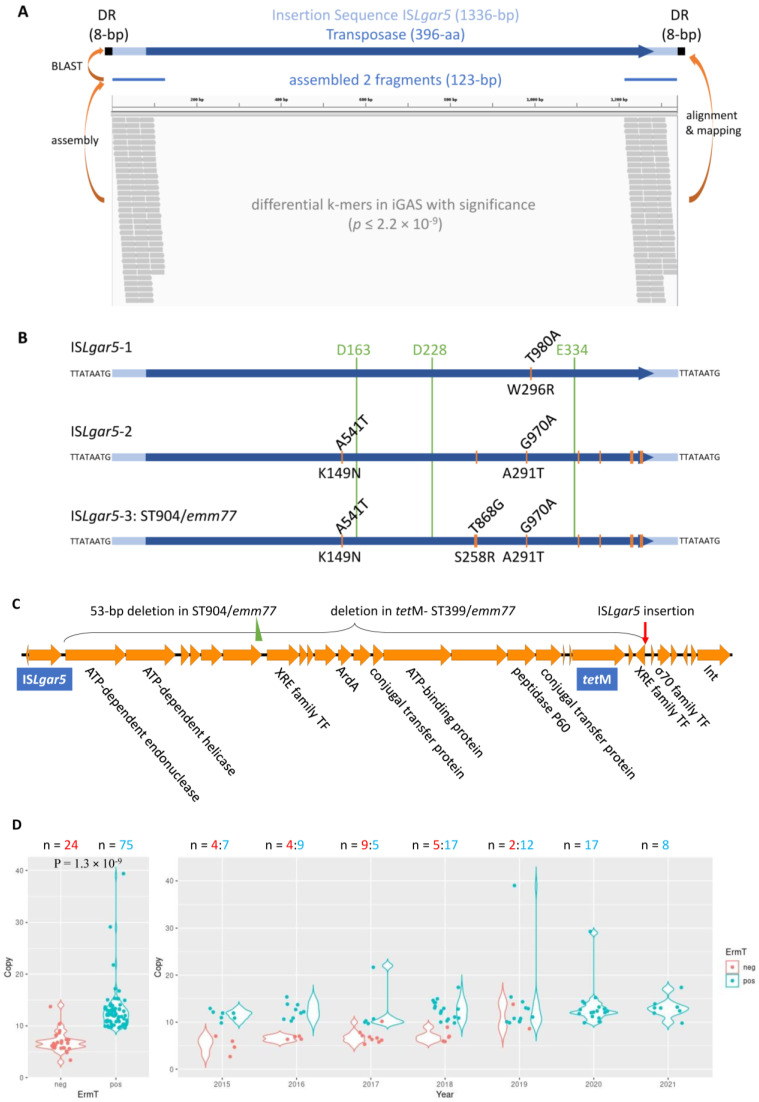
Identification of an insertion sequence, IS*Lgar5*, in association with invasive *Streptococcus pyogenes* antimicrobial resistance. (**A**) Identification of IS*Lgar5* in a genome-wide association study of Group A *Streptococcus* (GAS), though this was later found to be biased by multiple copies of IS*Lgar5* in ST399/*emm77* isolates. IS*Lgar5* encodes a transposase of 396 amino acids and features an 8 bp duplicate repeat (DR). (**B**) Three IS*Lgar5* variants identified in single-copy IS*Lgar5*-containing GAS, which has 1, 2, and 3 amino acids changed in its encoded transposase, respectively, as marked in the figure with nucleotide changes (orange bars) on the top and amino acid changes at the bottom. The DDE sites necessary for the transposase activity are also labeled. All single-copy IS*Lgar5* variants have the same conservative DR, TTATAATG. (**C**) A ~26 kb *Tn*5801-link integrative and conjugative element (ICE) carries both IS*Lgar5* and *tet*M. Gene annotation of the ICE is from emmSTG866.1, 986,768–960,966 in CP035428. Shown on the top are the 53 bp deletion in IS*Lgar5*-3-carrying ST904/*emm77* (green triangle), the deleted region in *tet*M-negative ST399/*emm77*, and the site of IS*Lgar5* insertion in some of the ICEs (red arrow). TF: transcription factor. (**D**) The abundance of IS*Lgar5* in ST399/*emm77* is associated with *erm*T antimicrobial gene. **Left**: Isolates without *erm*T (in red) have a much lower copy number of IS*Lgar5* than those with *erm*T (in cyan). **Right**: Isolates with *erm*T (in cyan) are increasing during the surveillance time, with more isolates without *erm*T (in red) in 2015–2018 (n = 21) than in 2019–2021 (n = 2).

**Figure 4 ijms-25-08179-f004:**
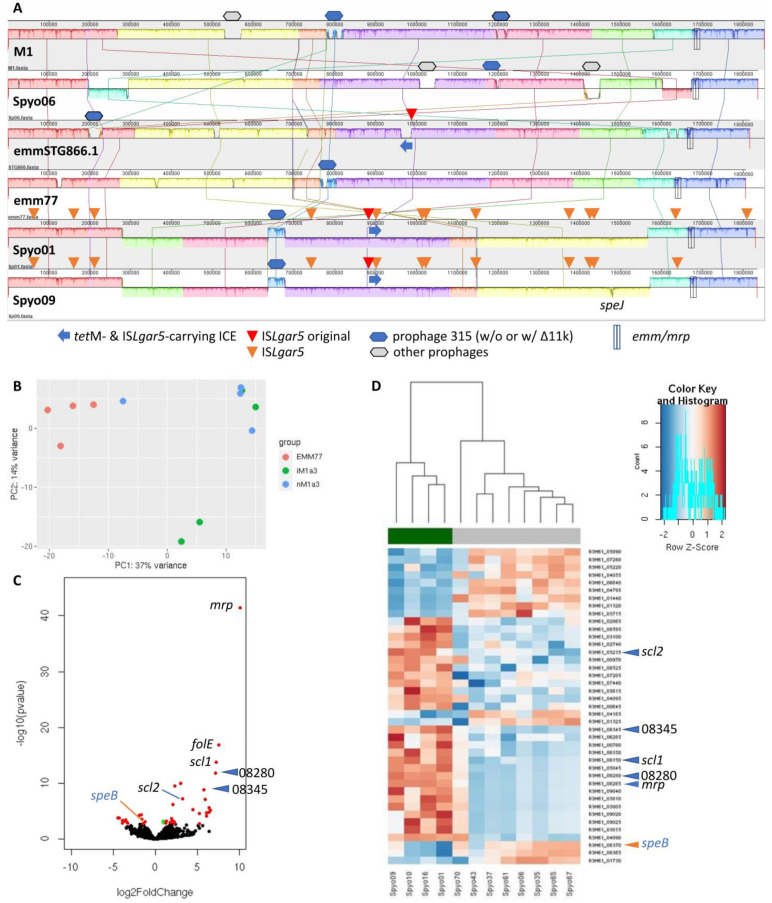
Genome-wide comparisons of Group A *Streptococcus* (GAS) chromosomal structure and gene expression. (**A**) Whole-genome sequence alignment of our new complete genomes, Spyo01 (CP136948), Spyo06 (CP136950), and Spyo09 (CP136951), along with reference genomes, M1 (AE004092), emm77 (CP035439), and emmSTG866.1 (CP035428). The original IS*Lgar5* in the *tet*M- and IS*Lgar5*-carrying ICE is shown with red triangle, compared to the diversified IS*Lgar5* throughout the chromosome, which is shown with orange triangle. (**B**) Principal component analysis of 12 GAS isolates in RNA sequencing, grouped into ST399/*emm77*, invasive M1a-3, and non-invasive M1a-3 (n = 4 each). (**C**) Volcano plot of differential gene expression between ST399/*emm77* (n = 4) and ST28/*emm1* (n = 8). Red dots represent 39 genes with significant changes (adjusted *p*-value ≤ 0.05 and |log_2_FoldChange| ≥ 1); green dots represent 2 genes with adjusted *p*-value ≤ 0.05 but |log_2_FoldChange| < 1. (**D**) Heatmap of genes differentially expressed between ST399/*emm77* (n = 4) and ST28/*emm1* (n = 8) with hierarchical clustering of these 12 GAS isolates. Genes associated with GAS virulence were marked in (**C**,**D**), including M protein-related *mrp* and R3H61_08280; LPXTG-anchored proteins, *scl1*, *scl2*, and R3H61_08345; and exotoxin gene *spe*B.

**Table 1 ijms-25-08179-t001:** Genotyping of local 95 Group A Streptococcus isolates in Eastern North Carolina during 2022–2023.

Genotype	iGAS		nGAS		uGAS		SUM		*p*-Value *
	n	%	n	%	n	%	n	%	
ST28/*emm1*	16	16.8	13	13.7	3	3.2	32	33.7	0.58
ST36/*emm12*	18	18.9	13	13.7	2	2.1	33	34.7	0.59
ST101/*emm89*	2	2.1	3	3.2	1	1.1	6	6.3	0.33
ST399/*emm77*	3	3.2			1	1.1	4	4.2	0.14
Others	15	15.8	3	3.2	2	2.1	20	21.1	0.11
SUM	54	56.8	32	33.7	9	9.5	95	100	

iGAS: invasive GAS; nGAS: non-invasive GAS; uGAS: urine-sourced GAS. *: Fisher’s exact probability test was used, comparing the distribution of one genotype to the combined distribution of others.

**Table 2 ijms-25-08179-t002:** National genotyping surveillance of invasive Group A Streptococcus in the U.S. during 2015–2021, with a total of 13,064 isolates in BioProject PRJNA395240. Shaded in blue: top one for the year; shaded in grey: the 2nd–5th ones for the year (from the darkest to the lightest).

MLST/*emm*	<2015	2015		2016		2017		2018		2019		2020		2021	
	n	n	%	n	%	n	%	n	%	n	%	n	%	n	%
ST28/*emm1*	1	301	20.7	236	12.9	312	13.4	336	13.9	359	15.1	195	10.5	17	2.7
ST101/*emm89*	2	176	12.1	151	8.3	232	9.9	181	7.5	139	5.9	115	6.2	28	4.5
ST36/*emm12*	3	132	9.1	149	8.1	137	5.9	128	5.3	136	5.7	76	4.1	6	1.0
ST334/*emm82*		94	6.5	106	5.8	133	5.7	159	6.6	159	6.7	133	7.1	78	12.5
ST82/*emm92*	2	57	3.9	55	3.0	166	7.1	263	10.9	170	7.2	125	6.7	50	8.0
ST433/*emm49*	117	19	1.3	155	8.5	184	7.9	137	5.7	112	4.7	123	6.6	95	15.3
ST3/*emm43*						21	0.9	62	2.6	41	1.7	47	2.5	31	5.0
ST909/*emm81*		19	1.3					3	0.1	7	0.3	37	2.0	29	4.7
ST399/*emm77*		11	0.8	13	0.7	14	0.6	22	0.9	14	0.6	17	0.9	8	1.3
Total	172	1454		1830		2333		2417		2374		1862		622	

## Data Availability

BioProject accession number: PRJNA1025265; BioSample accession numbers: SAMN37718901–SAMN37718995; SRA accession numbers for Illumina short-reads: SRR26338169–SRR26338263; SRA accession numbers for PacBio long-reads: SRR26324205–SRR26324207; Genome accession numbers: CP136948–CP136952; and GEO accession number: GSE246380.

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
