# Peer review of "Local Genomic Surveillance of Invasive Streptococcus pyogenes in Eastern North Carolina (ENC) in 2022–2023"

_ijms, 2024, doi:10.3390/ijms25158179_

Round 1

Reviewer 1 Report

Comments and Suggestions for Authors

This paper provides a comprehensive analysis of Streptococcus pyogenes (GAS) isolates at both local and national levels, using advanced genomic techniques to unveil significant insights into the pathogen's evolution and resistance mechanisms. However, a limitation of the paper is its lack of focus. The authors addressed too many research topics, but none was analyzed in depth.

The authors started with k-mer genotyping using the kWIP program. One limitation of kWIP is that it does not allow any statistical validation of the visual clustering of isolates. For statistical validation of the lineages and sublineages inferred from kWIP, additional tools or methods must be used. For instance, clustering algorithms that support bootstrapping or other forms of statistical tests are necessary to assess the robustness and reliability of the lineages and sub-lineages proposed in this study.

The authors pointed out that during the COVID-19 pandemic, the pattern of frequently isolated lineages changed. However, it is unclear whether this was a local or global change. Additionally, any possible associations between the COVID-19 pandemic and this genetic drift were not discussed. It is quite possible that it was just a coincidence between two independent events. The authors themselves mentioned the geographic differences in GAS circulation (lines 316-317).

Next, the authors focused on the distribution of ISLgar5 insertion sequences, which were present in multiple copies in several isolates. It was demonstrated that the isolates with multiple copies were characterized by the overexpression of several virulence genes. Again, it remains unclear whether the ISLgar5 copies affected the genes, or whether it was a gene copy number effect, or if the virulence strains lost control over the insertion sequence actively propagating in their genomes.

Remarkably, instead of a conclusion section, the manuscript ends with 'Limitations of the study', which were indeed numerous.

Specific comments:

  1. It seems to me that GWAS is not the proper term for the analysis performed in section 2.3. GWAS is a statistical approach that finds significant associations between genetic polymorphisms and phenotype. The authors used 'differential k-mer analysis' to identify insertion sequences. No attempts were made to verify statistically whether these inserts affect the phenotype of the isolates.
  2. What is ‘Zoom-in kWIP analysis’ in line 163? Is it a function of kWIP?
  3. 'tetM gene conferring resistance to tetracycline' (line 224), 'These erm genes are responsible for GAS resistance to erythromycin, azithromycin, clarithromycin, and clindamycin' (lines 245-246) – these statements must be supported by references to respective publications.
  4. Line 230, tetM should be in italic typeface.
  5. Lines 355-356: 'Our genomic surveillance also revealed the association of ISLgar5 with antimicrobial resistance.', as well as 'We noticed there is inconsistency between resistance phenotypes and genotypes' – it was not demonstrated as the authors did not check the antimicrobial resistance of isolates experimentally in this study.
  6. Lines 356-357: 'Additionally, our transcriptomics analysis suggested that multiple copy ISLgar5 might alter GAS virulence and pathogenesis.' – again, it was not shown that ISLgar5 regulates genes. It is possible that an invasive lifestyle required specific gene regulation and made ISLgar5 inserts unstable and prone to self-propagation.

I suggest a significant revision of the paper to make it more focused. I suggest starting from the conclusion that should contain a clear statement of what exactly the authors want to communicate to the readers.

Author Response

This paper provides a comprehensive analysis of Streptococcus pyogenes (GAS) isolates at both local and national levels, using advanced genomic techniques to unveil significant insights into the pathogen's evolution and resistance mechanisms. However, a limitation of the paper is its lack of focus. The authors addressed too many research topics, but none was analyzed in depth.

A: We appreciate the reviewer’s time and effort very much, as well as his positive comments on “comprehensive analysis” and “advanced genomic techniques”. Our manuscript focuses on two main topics, local and national iGAS surveillance and identification of ISLgar5 multiple copies in iGAS. In the surveillance, we showed the breadth with addition of CDC’s ABCs iGAS data publicly available and the depth with ST28/emm1 lineages and sub-lineages, from which we identified a novel sub-lineage in our ENC. The followed identification of multi-copy ISLgar5 represented additional depth of our genomic study, with the breadth of including a single-copy ISLgar5 survey. Since lots of genomic data were involved, we tried to provide a comprehensive analysis and overview to the readers.

The authors started with k-mer genotyping using the kWIP program. One limitation of kWIP is that it does not allow any statistical validation of the visual clustering of isolates. For statistical validation of the lineages and sublineages inferred from kWIP, additional tools or methods must be used. For instance, clustering algorithms that support bootstrapping or other forms of statistical tests are necessary to assess the robustness and reliability of the lineages and sub-lineages proposed in this study.

A:  We agree with the reviewer that kWIP itself does not include any statistical analysis of clustering. Since MDS was used in kWIP, we used a recently developed algorithm SigClust-MDS [48] to calculate the statistical significance of kWIP clustering. We have added this into the method part, as well as in the resulted Figures 1 and 2. We appreciate the reviewer’s constructive suggestion.

The authors pointed out that during the COVID-19 pandemic, the pattern of frequently isolated lineages changed. However, it is unclear whether this was a local or global change. Additionally, any possible associations between the COVID-19 pandemic and this genetic drift were not discussed. It is quite possible that it was just a coincidence between two independent events. The authors themselves mentioned the geographic differences in GAS circulation (lines 316-317).

A:  Our conclusion on iGAS pattern change during the COVID-19 pandemic is mainly from national surveillance data shown in Table 2. Our local surveillance data were limited in 2022-2023. As the reviewer suggested, we added a couple sentences to discuss the coincidence of these two events. However, the puzzle remains, since we know little about emm type of iGAS dominating the outbreak/circulation, besides the geographic differences.

Next, the authors focused on the distribution of ISLgar5 insertion sequences, which were present in multiple copies in several isolates. It was demonstrated that the isolates with multiple copies were characterized by the overexpression of several virulence genes. Again, it remains unclear whether the ISLgar5 copies affected the genes, or whether it was a gene copy number effect, or if the virulence strains lost control over the insertion sequence actively propagating in their genomes.

A:  It is more likely that under a certain selective pressure, e.g., antibiotics, the bacteria utilize transposition of an insertion sequence (or other mobile genetic elements), e.g., ISLgar5, for their own adaptation and genomic fitness. Multiple copies were generated during the bacterial evolution, as we showed in Figure 3. Our genomic sequencing data and RNAseq data only represented their final evolution result. A population-based evolution study may be required to address the reviewer’s question.

Remarkably, instead of a conclusion section, the manuscript ends with 'Limitations of the study', which were indeed numerous.

A: We have removed the section “Limitations of the study”, and incorporated them into the Discussion.

Specific comments:

It seems to me that GWAS is not the proper term for the analysis performed in section 2.3. GWAS is a statistical approach that finds significant associations between genetic polymorphisms and phenotype. The authors used 'differential k-mer analysis' to identify insertion sequences. No attempts were made to verify statistically whether these inserts affect the phenotype of the isolates.

A:  As a genetic element, k-mer has recently been adopted in GWAS analysis, as well as structural variants and others (DOI: 10.1016/j.tig.2021.05.006). Statistical significance, as shown in Figure 3A, has been used for the association study. Initially, we were trying to find k-mers or genes associated with invasive GAS. However, we found our association results were biased by the multiple copies of ISLgar5 in ST399/emm77, as we have discussed in the manuscript. Therefore, we did not go further with verification of the association.

What is ‘Zoom-in kWIP analysis’ in line 163? Is it a function of kWIP?

A:  “Zoom-in” here means centered or focused analysis. We have replaced the word in the revision to make it clearer.

'tetM gene conferring resistance to tetracycline' (line 224), 'These erm genes are responsible for GAS resistance to erythromycin, azithromycin, clarithromycin, and clindamycin' (lines 245-246) – these statements must be supported by references to respective publications.

A:  References added as suggested.

Line 230, tetM should be in italic typeface.

A:  Revised as suggested.

Lines 355-356: 'Our genomic surveillance also revealed the association of ISLgar5 with antimicrobial resistance.', as well as 'We noticed there is inconsistency between resistance phenotypes and genotypes' – it was not demonstrated as the authors did not check the antimicrobial resistance of isolates experimentally in this study.

A: The antimicrobial susceptibility tests are described in the methods and the results are shown in the supplemental Table S1.

Lines 356-357: 'Additionally, our transcriptomics analysis suggested that multiple copy ISLgar5 might alter GAS virulence and pathogenesis.' – again, it was not shown that ISLgar5 regulates genes. It is possible that an invasive lifestyle required specific gene regulation and made ISLgar5 inserts unstable and prone to self-propagation.

A:  As an insertion sequence, ISLgar5 itself may interfere other genes, but not regulate other genes.  Since ISLgar5 belongs to IS256 family and IS256 has been shown to have a tight regulatory control, it is reasonable for us to believe that ISLgar5 is stable too. However, under a certain circumstance (e.g., antibiotic selection), it may prone to self-propagation and transposition, leading to copy number increase, further altering GAS virulence and pathogenesis.

I suggest a significant revision of the paper to make it more focused. I suggest starting from the conclusion that should contain a clear statement of what exactly the authors want to communicate to the readers.

A:  We appreciate the reviewer’s constructive suggestions very much. We were aware that lots of genomic data were involved, and therefore used sub-titles in results for clear statements.

Reviewer 2 Report

Comments and Suggestions for Authors

The manuscript entitled “ Local and National Genomic Surveillance of Invasive Streptococcus pyogenes” describes a genomic surveillance activity that involved 95 GAS strains isolated locally and 13,064 strains isolated in USA.

I have major concerns over the findings of the study.

1-      Introduction: The paragraph is a little sparse. Please provide more references.

2-      Table 2: Please specify the color’s meaning.

3-      Line: 75: Please specify the number of isolates tested for antimicrobial susceptibility test.

4-      In results, the authors referred to genome length of the isolate, but what are the other assembly quality parameters obtained (n. contig, N50, etc).

5-      It is unclear why the authors preferred the k-mers method to test genomic correlation between strains.

6-      Discussion paragraph: The paragraph is a little sparse compared to the amount of data obtained and not very homogeneous. Please join the two parts.

Author Response

The manuscript entitled “Local and National Genomic Surveillance of Invasive Streptococcus pyogenes” describes a genomic surveillance activity that involved 95 GAS strains isolated locally and 13,064 strains isolated in USA.

I have major concerns over the findings of the study.

A:  We appreciate the reviewer’s time and effort very much.

  • Introduction:The paragraph is a little sparse. Please provide more references.

A:  We have revised the Introduction part as suggested.

  • Table 2: Please specify the color’s meaning.

A:  Added as suggested.

  • Line: 75: Please specify the number of isolates tested for antimicrobial susceptibility test.

A:  Added as suggested.

  • In results, the authors referred to genome length of the isolate, but what are the other assembly quality parameters obtained (n. contig, N50, etc).

A:  Added these contents in the supplemental table as suggested.

  • It is unclear why the authors preferred the k-mers method to test genomic correlation between strains.

A:  We have given our reasons of using k-mer method for our genomic surveillance in the last paragraph of the Introduction. We have discussed it as well in the paragraphs of the Discussion. Please refer to these paragraphs for the detail.

  • Discussion paragraph: The paragraph is a little sparse compared to the amount of data obtained and not very homogeneous. Please join the two parts.

A:  We have removed the part of “Limitations of the study”, incorporated these contents into the Discussion, and made some revisions significantly.

Reviewer 3 Report

Comments and Suggestions for Authors

Authors conducted a genomic surveillance, locally in eastern North Carolina (ENC) during 2022-2023 with 95 isolates of GAS and compared them to the U.S. iGAS in 2015-2021 that have been whole genome sequenced under the CDC’s Active Bacterial Core surveillance (ABCs) (13,064 isolates).

The work is of interest from an epidemiological point of view, allowing the monitoring of outbreaks, and providing data that can assist in the development of a vaccine.

I consider, however, that, taking into account that the original data in this study are only local, the title is somewhat abusive. It was not the authors who carried out the national research, they only worked on existing data. They should mention that it is a comparison of local surveillance with existing national data.

It also doesn't make much sense to compare data from different periods: local data from 2022-2023 with national data from 2015-2021.  An extremely small sample compared to a much larger one and in different periods, the bias is enormous. The comparisons drawn from the analyzes are already compromised.

It would be more interesting to make the comparison only with data from the same location, that is, from the national database to select only those from the same area, this way they could try to understand the local variation.

Genomic information is important, but I think it should be treated differently.

Author Response

Authors conducted a genomic surveillance, locally in eastern North Carolina (ENC) during 2022-2023 with 95 isolates of GAS and compared them to the U.S. iGAS in 2015-2021 that have been whole genome sequenced under the CDC’s Active Bacterial Core surveillance (ABCs) (13,064 isolates).

The work is of interest from an epidemiological point of view, allowing the monitoring of outbreaks, and providing data that can assist in the development of a vaccine.

A: We appreciate the reviewer’s time and effort.

I consider, however, that, taking into account that the original data in this study are only local, the title is somewhat abusive. It was not the authors who carried out the national research, they only worked on existing data. They should mention that it is a comparison of local surveillance with existing national data.

A: We appreciate the reviewer’s concern. We have revised the whole manuscript, including the title, abstract, and context, accordingly, and highlighted these changes.

It also doesn't make much sense to compare data from different periods: local data from 2022-2023 with national data from 2015-2021.  An extremely small sample compared to a much larger one and in different periods, the bias is enormous. The comparisons drawn from the analyzes are already compromised.

It would be more interesting to make the comparison only with data from the same location, that is, from the national database to select only those from the same area, this way they could try to understand the local variation.

Genomic information is important, but I think it should be treated differently.

A: As we have noted and discussed in our manuscript, this is a small data set (obviously due to funding issue), compared to a national big data set, using kWIP analysis; and there was a time frame difference. Notably, kWIP has not been applied into such a large genomic surveillance before, representing our effort on this novel comparative analyses, from which we revealed iGAS lineages, sub-lineages, and structural changes. We totally agree with the reviewer that it will be ideal to compare the surveillances with the same period and the same location, and we wish we had such ideal data sets. We regret that the national surveillance neither included our surveillance area nor had yet made the new data (2022-2023) publicly available. We are thus sorry that we cannot conduct the comparisons as the reviewer suggested. Nevertheless, as the reviewer may appreciate, through the comparisons we identified a new appearing sub-lineage in ENC; meanwhile, we are careful about any potential bias and have made our conclusions as less biased as possible in our genomic comparative surveillance.

Reviewer 4 Report

Comments and Suggestions for Authors

The manuscript titled “Local Genomic Surveillance of Invasive Streptococcus pyogenes in Eastern North Carolina (ENC) 2022-2023” presents a comprehensive genomic surveillance study focusing on Group A Streptococcus, specifically its invasive strains in Eastern North Carolina. The authors compare their findings with the 13,064 isolates from the 2015-2021 USA National Genomic Surveillance using the k-mer approach.

The study is well-designed, and the results are presented in a manner that robustly supports the conclusions.

To further enhance the manuscript, please consider the following minor suggestions:

  1. Line 69: Please add "(iGAS)" after “Invasive GAS” for clarity. The term "iGAS" is used in line 74 without prior definition.
  2. Reagents and Equipment: Throughout the manuscript, specify the manufacturer, city, and country for all mentioned reagents and equipment.
  3. Conclusion Section: Although optional, consider adding a conclusion paragraph at the end of the discussion section to summarize the key findings and implications of your study.

These adjustments will improve the clarity and comprehensiveness of the manuscript.

Author Response

Line 69: Please add "(iGAS)" after “Invasive GAS” for clarity. The term "iGAS" is used in line 74 without prior definition.

The term iGAS was used in Line 34 before Lines 69 and 74. Since the sentence started with iGAS, we thought it would be better using "invasive GAS" than "iGAS". Thanks for the note. 

Reagents and Equipment: Throughout the manuscript, specify the manufacturer, city, and country for all mentioned reagents and equipment.

We appreciate the reviewer's note, and have revised the Material and Methods section accordingly. Please refer to the highlighted parts in the revised manuscript. 

Conclusion Section: Although optional, consider adding a conclusion paragraph at the end of the discussion section to summarize the key findings and implications of your study.

We appreciate the reviewer's constructive suggestion, and have added a section as suggested. Many thanks. 

Round 2

Reviewer 1 Report

Comments and Suggestions for Authors

I consider this work comprehensive, but a comprehensive paper needs a solid conclusion. This is the problem. I appreciate the authors' attempt to improve their manuscript, but it is still far from acceptable.

The paper is built of 3 separate research stories, which do not work together:

1.       Comparative genomics of 95 new isolates of Streptococcus pyogenes;

2.       Identification of novel sub-lineages of the most abundant lineage ST28/emm1;

3.       Analysis of distribution of ISLgar5 repetitive elements in different genome, with an attempt to make biological senses based on these differences.

Each story could be a separate paper, but none are finalized with a proper conclusion, nor do they lead to a general conclusion.

The major drawbacks are:

1.       iGAS and nGAS are not properly defined. It is said that: “Invasive GAS was defined as the collection of a GAS isolate from a normally sterile site or from a wound in a patient with necrotizing fasciitis or streptococcal toxic shock syndrome (STSS), according to the standard [18].”

a.       I did not find in the Methods how STSS was detected. Was it an experimental procedure?

b.       Add columns to the supplementary tables with explicit associations of isolates with iGAS/nGAS. I managed to associate them by inspecting formulas in Excel spreadsheets, but it seemed like a random distribution.

c.       Fix the typing error: “fasciitis”

2.       The tree in Figure 1 was introduced (line 91) as a ‘phylogenetic tree,’ implying that organisms with recent common ancestry were grouped together. However, it doesn’t appear so. While clustering is good, the clusters are polyphyletic. Strains belonging to different MLST types, isolated from different sources, and with different phenotypes were grouped together (see Suppl. Table 2). The authors acknowledge this (Ln. 88-89): “No specific genotype was found to be significantly associated with iGAS infection (Table 1).”
This is a very important question that potentially leads to a cornerstone conclusion regarding GAS biology. If this is truly a phylogenetic tree, we may conclude that virulence and infection site adaptation in GAS evolve in parallel across multiple lineages, independently of their phylogenetic relations. This contradicts our general knowledge about many other pathogens. For instance, uropathogenic E. coli evolved from a phylogenetically separated lineage. To make such a fundamental conclusion regarding GAS uniqueness, the authors need to perform additional phylogenetic studies to confirm that the tree in Fig. 1 is indeed a phylogenetic tree.
If it is not a phylogenetic tree, what is it and what is it useful for? The authors advocate the k-mer approach as annotation independent. While true, 31-mer matches are biased by sequencing and assembly errors and by the abundance of repetitive elements, such as ISLgar5. Most likely, this approach clustered genomes based on different frequencies of ISLgar5. If this is true, it clearly demonstrates that ISLgar5 repeats are irrelevant to GAS virulence and phenotype.

3.       Regarding the second story on the identification of novel sub-lineages.

a.       How were the plots in Fig. 2 calculated? This is not explained. Is it kWIP? In line 354, it is stated: “We employed both k-mer-based relatedness and differential analyses to investigate ST28/emm1”. What are these “differential analyses”? If kWIP is used, we return to my previous question: is this phylogeny or random clustering based on different numbers of meaningless repetitive transposons?

b.       What do the authors want to conclude from this analysis? Only that the GAS population is genetically versatile? It would be great if the authors linked this versatility to some evolutionary trends, disease outbreaks, climate change, or other factors to make some biological or medical sense. However, this would require a more detailed analysis of metadata associated with different isolates, which might justify a separate publication.

4.       The third story on ISLgar5 is the weakest.

a.       The quality of genome assembly resulting in multiple short contigs (Suppl. Table 2) based on Illumina short reads does not allow for precise counting of ~1,400 bp repetitive elements. Some were likely counted multiple times, while others were missed.

b.       The authors attempted to associate the number of repeats with the number of virulence or antibiotic resistance genes. I see no statistics to support this conclusion.

c.       The authors decided to sequence two representative strains belonging to different MLST types using PacBio, which was a good idea. They found that one strain has 14 copies of ISLgar5 and another genome has 13 copies. Not a significant difference. Still, the authors groundlessly insist that this difference can explain the observed differences in gene expression profiles (Fig. 4). This is absolutely wrong and misleading. Many other factors, such as SNPs in promoter regions, epigenetic modifications, and alternative chromosome conformations, can lead to gene expression differences. An extra copy of a transposon is the least likely reason.  

5.       Discussion, ln. 334-336: “Our local genomic surveillance of GAS revealed a unique sub-lineage, M1a-5, in ENC among the most common iGAS ST28/emm1. However, it was conducted in a relatively short time period in a relatively small size and was confined to ENC, a relatively large geographic region with a relatively small population.” Can we replace the four instances of “relatively” with the fact that the authors do not have enough data to make a statistically reliable conclusion?

6.       Generally, the paper is difficult to read and follow due to the abundance of unimportant facts, which the authors found “interesting,” but which lead to no conclusions. There is no need to include all observations in the paper. Only keep facts that support the main conclusions. Keep the irrelevant “interesting facts” for future studies and publications.  

Author Response

I consider this work comprehensive, but a comprehensive paper needs a solid conclusion. This is the problem. I appreciate the authors' attempt to improve their manuscript, but it is still far from acceptable.

The paper is built of 3 separate research stories, which do not work together:

  1. Comparative genomics of 95 new isolates of Streptococcus pyogenes;
  2. Identification of novel sub-lineages of the most abundant lineage ST28/emm1;
  3. Analysis of distribution of ISLgar5 repetitive elements in different genome, with an attempt to make biological senses based on these differences.

Each story could be a separate paper, but none are finalized with a proper conclusion, nor do they lead to a general conclusion.

A:  We appreciate the reviewer’s opinions. However, we do think these research stories are continuously connected since ISLgar5 was identified during the genomic survey. Separating these stories into individual manuscripts would certainly break them into pieces and weaken the whole survey story. Notably, ISLgar5 repeats only occur in ST399/emm77 except one isolate of ST458/EMM-28. ISLgar5 repeats do not occur in different genomes, although single-copy ISLgar5 does occur in some other genomes. This is one of the solid conclusions in our manuscript.

The major drawbacks are:

  1. iGAS and nGAS are not properly defined. It is said that: “Invasive GAS was defined as the collection of a GAS isolate from a normally sterile site or from a wound in a patient with necrotizing fasciitis or streptococcal toxic shock syndrome (STSS), according to the standard [18].”
  2. I did not find in the Methods how STSS was detected. Was it an experimental procedure?
  3. Add columns to the supplementary tables with explicit associations of isolates with iGAS/nGAS. I managed to associate them by inspecting formulas in Excel spreadsheets, but it seemed like a random distribution.
  4. Fix the typing error: “fasciitis”

A:  We added a column in the supplementary Table S1 to clarify iGAS/nGAS isolates. STSS is a clinical definition according to CDC (https://ndc.services.cdc.gov/case-definitions/streptococcal-toxic-shock-syndrome-2010/), not from any experimental procedure. The standardized definition of iGAS is referenced correctly; and “fasciitis” is spelled correctly.

  1. The tree in Figure 1 was introduced (line 91) as a ‘phylogenetic tree,’ implying that organisms with recent common ancestry were grouped together. However, it doesn’t appear so. While clustering is good, the clusters are polyphyletic. Strains belonging to different MLST types, isolated from different sources, and with different phenotypes were grouped together (see Suppl. Table 2). The authors acknowledge this (Ln. 88-89): “No specific genotype was found to be significantly associated with iGAS infection (Table 1).”
    This is a very important question that potentially leads to a cornerstone conclusion regarding GAS biology. If this is truly a phylogenetic tree, we may conclude that virulence and infection site adaptation in GAS evolve in parallel across multiple lineages, independently of their phylogenetic relations. This contradicts our general knowledge about many other pathogens. For instance, uropathogenic E. coli evolved from a phylogenetically separated lineage. To make such a fundamental conclusion regarding GAS uniqueness, the authors need to perform additional phylogenetic studies to confirm that the tree in Fig. 1 is indeed a phylogenetic tree.
    If it is not a phylogenetic tree, what is it and what is it useful for? The authors advocate the k-mer approach as annotation independent. While true, 31-mer matches are biased by sequencing and assembly errors and by the abundance of repetitive elements, such as ISLgar5. Most likely, this approach clustered genomes based on different frequencies of ISLgar5. If this is true, it clearly demonstrates that ISLgar5 repeats are irrelevant to GAS virulence and phenotype.

A: Figure 1 is a genome-based phylogenetic tree using the kWIP method, comparable to phylogenetic trees using SNP analysis shown in Figure S1 and using core-genome analysis shown in Figure S2. Notably, in all these genome-wide phylogenetic trees, ST28/emm1 isolates are clustered together, so are ST36/emm12 isolates. Focused on other STs, the same ST/emm isolates are also clustered together, demonstrating their evolutionary closeness. Based upon these comparisons, we conclude that kWIP analysis has higher resolution than SNP and core-genome analyses have. This is another solid conclusion in our manuscript.

    Our conclusion that “No specific genotype was found to be significantly associated with iGAS infection” was data-based with statistical p-values shown in Table 1, and thus was evidence-based, at least for our data set. As we discussed in our manuscript, our dataset is relatively small. To make a more solid conclusion on this, a large-scale survey is needed. Notably, invasive phenotype of GAS is unique, not comparable to uropathogenic E. coli or other pathogens.

    As we noted above, ISLgar5 repeats only occur in ST399/emm77 except one isolate of ST458/EMM-28. ISLgar5 repeats do not occur in different genomes, although single-copy ISLgar5 does occur in some other genomes. We appreciate the reviewer’s note that k-mer analysis may be “biased by sequencing and assembly errors and by the abundance of repetitive elements”. Notably, genome-wide phylogenetic analysis and genome-wide association study (GWAS) are two different analyses. As parts of a whole genome, repetitive elements should have been included in genome-wide phylogenetic analysis. However, heavy repetitive elements might bring potential bias to the GWAS. We used genome assembly for both kWIP phylogenetic analysis and GWAS. Genome assembly of Illumina short reads usually generates only one copy of any long repetitive elements, increasing bias for phylogenetic analysis but reducing bias in GWAS. Nevertheless, the edges of repetitive elements are still informative and useful, being detected in the kmdiff differential analysis, as shown in Figure 3A.

    We drew the conclusion on ISLgar5 copy number associated with antibiotic resistance (ermT) in ST399/emm77, as shown in Figure 3D, one of solid conclusions in our manuscript. As we clarified in our previous response, due to the potential bias of repetitive elements in the GWAS analysis, we did not go further with ISLgar5 association with invasive GAS phenotype.

  1. Regarding the second story on the identification of novel sub-lineages.
  2. How were the plots in Fig. 2 calculated? This is not explained. Is it kWIP? In line 354, it is stated: “We employed both k-mer-based relatedness and differential analyses to investigate ST28/emm1”. What are these “differential analyses”? If kWIP is used, we return to my previous question: is this phylogeny or random clustering based on different numbers of meaningless repetitive transposons?
  3. What do the authors want to conclude from this analysis? Only that the GAS population is genetically versatile? It would be great if the authors linked this versatility to some evolutionary trends, disease outbreaks, climate change, or other factors to make some biological or medical sense. However, this would require a more detailed analysis of metadata associated with different isolates, which might justify a separate publication.

A:  Yes, Figure 2 is a kWIP analysis on local and national ST28/emm1 isolates. The statement in lines 354-355 is regarding ISLgar5 identification, using kWIP and kmdiff analyses. We have revised both Figure 2 legend and the statement in lines 354-355 to clarify. As we noted above, in kWIP analysis with the assembled genomes, any long repetitive elements had been reduced to a minimum because of genome assembly.

     What we found in this analysis are the sub-lineages of ST28/emm1 (genetic versatility) and their changes during the surveillance years, shown in the tables of Figure 2. We appreciate the reviewer’s note that detailed associations with “some evolutionary trends, disease outbreaks, climate changes, or other factors” require detailed “metadata associated with different isolates”. We regret that currently, there is no such metadata publicly available, hindering us from further analysis.

  1. The third story on ISLgar5 is the weakest.
  2. The quality of genome assembly resulting in multiple short contigs (Suppl. Table 2) based on Illumina short reads does not allow for precise counting of ~1,400 bp repetitive elements. Some were likely counted multiple times, while others were missed.
  3. The authors attempted to associate the number of repeats with the number of virulence or antibiotic resistance genes. I see no statistics to support this conclusion.
  4. The authors decided to sequence two representative strains belonging to different MLST types using PacBio, which was a good idea. They found that one strain has 14 copies of ISLgar5 and another genome has 13 copies. Not a significant difference. Still, the authors groundlessly insist that this difference can explain the observed differences in gene expression profiles (Fig. 4). This is absolutely wrong and misleading. Many other factors, such as SNPs in promoter regions, epigenetic modifications, and alternative chromosome conformations, can lead to gene expression differences. An extra copy of a transposon is the least likely reason.  

A:  The genome assembly of Illumina short reads usually generates only one copy of any long repetitive elements, not allowing for their precision counting. As described in our manuscript (lines 210-212), we used sequencing depth ratio to calculate the number of repeats, the depth on repeats vs. the average depth on the genome. The association of ISLgar5 copy number with antibiotic resistance (ermT) in ST399/emm77 was shown in Figure 3D, which represents another solid conclusion of our study.

    The differential gene expression shown in Figure 4 were from the comparison of ST399/emm77 to ST28/emm1, not from the comparison of 14-copy isolate to 13-copy isolate. We have no statement in the manuscript or intention to mislead the readers that the differential gene expression was solely caused by ISLgar5 multiple copies.

  1. Discussion, ln. 334-336: “Our local genomic surveillance of GAS revealed a unique sub-lineage, M1a-5, in ENC among the most common iGAS ST28/emm1. However, it was conducted in a relatively shorttime period in a relatively small size and was confined to ENC, a relatively large geographic region with a relatively small population.” Can we replace the four instances of “relatively” with the fact that the authors do not have enough data to make a statistically reliable conclusion?

A:  The conclusion on the unique sub-lineage M1a-5 in ENC is a solid fact, supported by Figure 2C. Our four instances of “relatively” were in regard to our local surveillance, nothing related to statistical reliability on the conclusion. We have revised the sentence accordingly.

  1. Generally, the paper is difficult to read and follow due to the abundance of unimportant facts, which the authors found “interesting,” but which lead to no conclusions. There is no need to include all observations in the paper. Only keep facts that support the main conclusions. Keep the irrelevant “interesting facts” for future studies and publications.  

A:  We respectfully disagree with the reviewer’s opinions on separating our manuscript. We would like to present the whole story together.

Reviewer 2 Report

Comments and Suggestions for Authors

Thanks for your response.

Author Response

We appreciate the reviewer's time and effort very much.

Reviewer 3 Report

Comments and Suggestions for Authors I understand your limitations, it is certainly a pity that you are not able to have more recent data and that there are no data relating to your area in the national database. I believe that the changes carried out make the article more realistic. You have the opportunity to contribute with your data to the national database.

Author Response

We appreciate very much the positive response from the reviewer. Many thanks, 

Round 3

Reviewer 1 Report

Comments and Suggestions for Authors

I appreciate the amount of data generated by the authors but was not convinced by their replies to my comments. I suggest performing a more detailed analysis of the data, making the paper more focused and properly discussed. It will likely be necessary to split the current manuscript into 2-3 separate papers or remove a significant part of the current content. I do not think that kWIP is a good tool for phylogenetics as it does not follow any evolutionary theory. The figure provided by the authors reinforced my opinion. I am not convinced that the number of transposon inserts in a genome can influence the phenotype just because of the copy numbers. The phenotype may be affected only if the transposon copy appeared inside a functional gene or brought new genes. I found the associations between copy numbers of the inserts and the phenotype to be incorrect and misleading. Therefore, I cannot recommend this manuscript for publication.

Author Response

I appreciate the amount of data generated by the authors but was not convinced by their replies to my comments. I suggest performing a more detailed analysis of the data, making the paper more focused and properly discussed. It will likely be necessary to split the current manuscript into 2-3 separate papers or remove a significant part of the current content.

A: We regret that the reviewer has a different view of our manuscript. As we replied before, we would like to keep our story intact and hold them into one piece. We thank the reviewer’s time and effort.

I do not think that kWIP is a good tool for phylogenetics as it does not follow any evolutionary theory. The figure provided by the authors reinforced my opinion.

A: We think kWIP is better than SNV analysis and core-/pan-genome analysis and provided reasons in the introduction and data-proven evidence (shown in figures for comparisons). We regret that the reviewer has an interpretation of our figures different from ours.

I am not convinced that the number of transposon inserts in a genome can influence the phenotype just because of the copy numbers. The phenotype may be affected only if the transposon copy appeared inside a functional gene or brought new genes. I found the associations between copy numbers of the inserts and the phenotype to be incorrect and misleading. Therefore, I cannot recommend this manuscript for publication.

A: We believe most of transposons are found in intergenic regions, which may change the gene expression regulation, in addition to being in a functional gene. While some transposons harbor additional genes, some transposons do not. The phenotype is truly a reflection of genome-wide gene expression, which can be determined by mono-gene, but mostly by poly-genes. Transposon copies can change the genome structure, as well as gene regulations, including epigenomic regulation. To date, we have less knowledge about this field and we think this field deserves further investigation.